# Differential Adjusted Parity for Learning Fair Representations

**Bucher Sahyouni**
University of Surrey
Guildford, United Kingdom
bs00826@surrey.ac.uk

**Matthew Vowels**
The Sense, CHUV
Lausanne, Switzerland
Kivira Health
New York, NY, USA
matthew.vowels@unil.ch

**Liqun Chen**
University of Surrey
Guildford, United Kingdom
liqun.chen@surrey.ac.uk

**Simon Hadfield**
University of Surrey
Guildford, United Kingdom
s.hadfield@surrey.ac.uk

## Abstract

We introduce the Differential Adjusted Parity (DAP) loss to produce unbiased informative representations. DAP utilises a differentiable variant of the adjusted parity metric to create a unified objective function that combines downstream task classification accuracy and its inconsistency across sensitive feature domains. A key element in this approach is the use of Soft Balanced Accuracy, which makes the metric differentiable while remaining suitable for imbalanced problems. In contrast to previous non-adversarial approaches, DAP does not suffer a degeneracy where the metric is satisfied by performing equally poorly across all sensitive domains. On Adult and COMPAS, DAP outperforms several adversarial models on downstream task accuracy and fairness. The largest gains reach 22.5%, 44.1% and 40.1% on demographic parity, equalized odds and sensitive feature accuracy, respectively, when compared to the best performing adversarial approach for each metric.

## 1 Introduction

As artificial intelligence (AI) and machine learning (ML) models become increasingly prevalent, the need for responsible, fair, and unbiased machine learning is critical (Barocas & Selbst, 2016). Learned representations distill complex data into compressed forms that capture key patterns for efficient learning and prediction (Bengio et al., 2013), but they may also encode sensitive attributes like gender or race, thus leading to biased decisions that can perpetuate societal inequities.

Debiasing methods aim to remove sensitive information while maintaining task performance. Adversarial training has become popular, introducing an adversarial model during training to learn sensitive features, but this min-max game can be unstable and computationally intensive (Arjovsky et al., 2017; Zhang et al., 2018). Non-adversarial approaches offer greatly improved training stability, but they can struggle to achieve good task performance and may satisfy the loss by performing equally poorly across all sensitive domains. Representative adversarial approaches include ALFR, CFair and LAFTR (Edwards & Storkey, 2016; Zhao et al., 2020; Madras et al., 2018); representative non-adversarial approaches include FBC, FRC, BFA and VFAE (Gitiaux & Rangwala, 2021; Quan et al., 2022; 2023; Louizos et al., 2016).

We therefore propose a differentiable variant of the adjusted parity metric as a single objective function that considers both task classification accuracy and its invariance across sensitive feature domains, without an adversarial component. Our technique extends to multi-class problems, ensuring fairness across various sensitive features. A key element is the use of Soft Balanced Accuracy, which makes the objective smoothly differentiable and suitable for imbalanced problems. Unlike previous non-adversarial approaches, the metric cannot be satisfied by performing equally poorly

across all sensitive domains. Across Adult and COMPAS, DAP consistently improves adjusted parity and standard fairness metrics over NODEBIAS (reference network devoid of fairness constraints) and adversarial baselines.

## 2 METHOD

Fairness metrics such as demographic parity and equalized odds remain important for evaluation (Dwork et al., 2012; Hardt et al., 2016a;b), but they do not themselves provide a differentiable training objective that also rewards end-task performance. As an initial step towards solving this issue, Vowels et al. (2020) introduced a parity metric for evaluating domain invariance. We extend that adjusted parity metric to an arbitrary number of sensitive domains:

$$\Delta_{\text{adj}} = \frac{\bar{S} - S^R}{1 - S^R} \left(1 - \frac{\sigma}{\gamma}\right), \tag{1}$$

where $S^R$ is the baseline accuracy of a random predictor, $\bar{S}$ is the average accuracy across sensitive domains, $\sigma$ is the standard deviation across these domains and $\gamma$ is the maximum standard deviation. This normalises the metric between $[0, 1]$. For any even number of domains $(N)$ the value remains at $\gamma = 0.5$. However for an odd number of domains:

$$\gamma = \sqrt{\frac{1}{4}\left(1 - \frac{1}{N^2}\right)}. \tag{2}$$

The full derivation can be found in the appendix.

To obtain a differentiable variant of the adjusted parity metric, we replace the hard accuracy term with "Soft Balanced Accuracy". The soft accuracy is computed by omitting the "argmax" function from a standard classification accuracy metric. For predicted class probabilities $P(x)$ and one-hot encoded labels $L_x$, the vector of soft True Positives for all classes is

$$\text{TP} = \sum_x P(x) \odot L_x, \tag{3}$$

and the vector of soft False Negatives is

$$\text{FN} = \sum_x (1 - P(x)) \odot L_x, \tag{4}$$

where $\odot$ represents the Hadamard product. We then compute the Soft Balanced Accuracy as the average of the per-class recall:

$$S = \frac{1}{C} \left| \frac{\text{TP}}{(\text{TP} + \text{FN})} \right|^1, \tag{5}$$

where $C$ is the number of classes and $|.|^1$ represents the L1 norm. This gives equal weighting to all classes irrespective of their prevalence in the dataset.

We compute this soft balanced accuracy independently on subsets of the dataset corresponding to each sensitive domain. The mean and standard deviation of $S$ across these domains are then substituted for $\bar{S}$ and $\sigma$ in Equation 1, yielding the Differential Adjusted Parity (DAP) loss used for training. The resulting Differential Adjusted Parity gives a single cooperative objective that encourages improvements in task accuracy across all labels while penalizing inconsistency across sensitive domains. In our experiments, DAP is combined with the standard task cross-entropy loss, $\mathcal{L}_{ce}$, while $\beta$ controls the contribution of the standard deviation term and $\Omega$ controls the contribution of $\mathcal{L}_{ce}$. Intuitively, minimising task prediction inconsistency across sensitive domains would minimise the mutual information between the representations and the sensitive feature. Any classifier that exhibits either minimal consistency or accuracy yields $\Delta_{\text{adj}} = 0$, and unlike previous non-adversarial approaches the metric cannot be satisfied by performing equally poorly across all sensitive domains.

## 3 EXPERIMENTS

**Setup.** We evaluate on two widely acknowledged fairness benchmarks: Adult and COMPAS (Dua & Graff, 2017; Dieterich et al., 2016). Adult is a binary income classification task, where we use

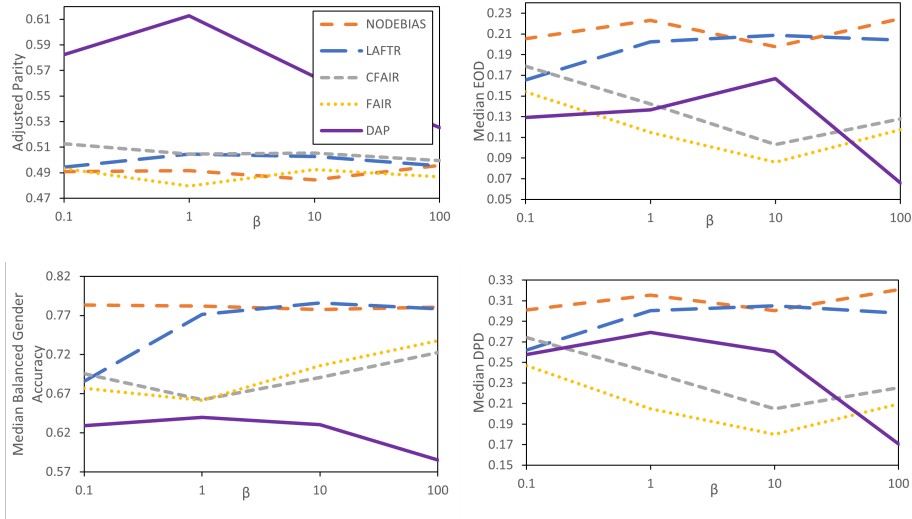

Figure 1: Comparing performance and fairness of all 4 models against DAP on Adult dataset. Top-left, top-right, bottom-left and bottom-right graph show how adjusted parity, equalised odds difference (EOD), gender classification accuracy and demographic parity difference (DPD) change with $\beta$. Higher adjusted parity and lower EOD, DPD and gender accuracy are favourable. DAP has higher adjusted parity and lower gender classification accuracy at all $\beta$. Lowest EOD and DPD are obtained by DAP at $\beta$=100.

gender as the sensitive feature. COMPAS is a recidivism prediction task, where we use race as the sensitive feature. We compare DAP against NODEBIAS, FAIR (ALFR), CFair and LAFTR, using the same evaluation protocol and implementation adapted from Kim (2021). For evaluation, we train two balanced random forest classifiers to predict the target and sensitive features from the learned encodings, and report adjusted parity, equalised odds difference (EOD), demographic parity difference (DPD), and sensitive-feature balanced accuracy. Models are trained for 20 epochs, sweeping $\Omega \in \{0...100\}$ and $\beta \in \{0.1...100\}$, and we report median and standard deviation over 5 runs. Full preprocessing, split details and implementation details are moved to the appendix.

**Results.** As depicted in Figure 1, DAP consistently outperforms NODEBIAS, LAFTR, CFAIR, and FAIR in terms of adjusted parity and gender classification balanced accuracy on the Adult dataset. When benchmarked against EO and DP metrics, DAP either achieves equivalent or better than the other 4 models. The lowest EOD and DPD are achieved with DAP. With $\beta = 100$, DAP obtains 44.1% and 18.6% lower EOD and DPD than the next best performer FAIR.

On the COMPAS dataset, DAP exhibits consistent improvements over previous state-of-the-art, particularly for high values of $\beta$. DAP achieves an adjusted parity that is improved by 45.9% and an EOD that is reduced by 12.4% when compared with its nearest competitor, CFAIR. Similarly, it registers a 22.5% improvement in DPD and a substantial 40.1% reduction in race classification accuracy when compared to FAIR, the latter being the second-best performer for these metrics. Additional sensitivity analyses, balanced versus unbalanced training, fairness–utility trade-off plots, and the multi-class sensitive-feature setting are moved to the appendix.

## 4 CONCLUSION

We formulate a differentiable variant of the adjusted parity metric for learning fair representations. By using Soft Balanced Accuracy and a cooperative, non-adversarial objective, DAP improves task performance consistency across sensitive domains without the instability of adversarial training. Across Adult and COMPAS, DAP outperforms established adversarial baselines on adjusted parity and standard fairness metrics. A limitation is sensitivity to the hyperparameters $\beta$ and $\Omega$, motivating future work on automatic calibration and adaptive weighting.

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

# A    BACKGROUND AND RELATED WORK

Below we will first discuss modern adversarial techniques for debiasing. We will then cover the non-adversarial techniques which are more closely related to this paper. This is followed by a formalisation of the various definitions of fairness found in the literature.

## A.1    ADVERSARIAL DEBIASING

Adversarially Learned Fair Representations, ALFR, is one of the earliest models developed to reduce bias and learn fair informative representations. It utilises an adversary which tries to predict the sensitive feature from the representations (Edwards & Storkey, 2016). The autoencoder tries to make these predictions as difficult as possible. It employs task and reconstruction losses to ensure the representation are informative for the downstream task, and an additional sensitive feature loss to remove sensitive feature related information.

The Conditional Fair Representations, CFair, approach stands as a seminal method aiming for accuracy parity (Zhao et al., 2020). Within the confines of conventional fair adversarial networks, CFair augments the original adversarial constraints and adopts conditional error constraints.

Learning Adversarially Fair and Transferable Representation, LAFTR, is similar to CFAIR but it uses one adversary instead of two and an L1 instead of a cross entropy loss to debias the representations (Madras et al., 2018). It still utilises a global cross-entropy loss for the target variable.

## A.2    NON-ADVERSARIAL DEBIASING

Outside of adversarial learning, Fairness by Compression (FBC) advocates for the use of binary compression to mitigate sensitive elements in representations (Gitiaux & Rangwala, 2021). They establish that the cross-entropy between $P(z)$ and $Q$ stands as the upper bound for the entropy $H(z)$. In this context, $P(z)$ delineates the distribution of a factorized representation, and $Q$ is utilized to predict $z_i$ based on $\{z_0, z_1, \ldots, z_{i-1}\}$. The FRC model aims to mitigate the influence of sensitive factors in the data representation by adjusting the correlation between the representation and the sensitive vectors (Quan et al., 2022).

BFA draws inspiration from the correlation coefficient constraints used in FRC (Quan et al., 2023). The primary ambition is to minimize the correlation between sensitive information and prediction error (as opposed to minimising correlation between sensitive information and representations in FRC), aiming to maintain high predictive accuracy.

The Variational Fair Autoencoder, VFAE, comprises of a variational autoencoder (VAE) instead of an autoencoder and employs an addititional maximum mean discrepancy (MMD) loss to ensure less sensitive information, which may be correlated to the target task, leaks into the learned representation (Louizos et al., 2016). MMD minimises the mismatch in moments between the marginal posterior distributions for the different sensitive features.

These techniques exhibit greatly improved training stability compared to the adversarial training approaches. However, they struggle to achieve good task performance (i.e. developing useful representations). It can be observed that many of these techniques experience a degeneracy where the loss function can be satisfied by performing equally poorly across all sensitive domains. In contrast our proposed approach maintains the benefits of non-adversarial training, while removing this degeneracy.

## A.3    MEASURES OF FAIRNESS

Fairness in machine learning has been extensively studied, and there exist a range of metrics which measure different aspects of it. Our primary contribution is the introduction of a new differentiable fairness metric, thus necessitating a brief overview of commonly used metrics in our model evaluation.

Demographic parity, also known as statistical parity or group fairness, requires that the selection rate (the rate at which individuals are positively classified) should be the same across all demographic groups. Mathematically, if $Y$ is the predicted label and $A$ denotes the demographic group,

demographic parity is defined as:

$$P(Y = 1|A = 0) = P(Y = 1|A = 1). \tag{6}$$

This implies that the algorithm should be independent of the sensitive attribute $A$, which can be limiting if the attribute is relevant to the outcome (Dwork et al., 2012; Hardt et al., 2016b).

The equalized odds fairness metric demands true positive rates and false positive rates to be equal across demographic groups. Mathematically, if $\hat{Y}$ is the true label:

$$P(Y=1|\hat{Y}=1, A=0) = P(Y=1|\hat{Y}=1, A=1) \tag{7}$$

and

$$P(Y=1|\hat{Y}=0, A=0) = P(Y=1|\hat{Y}=0, A=1). \tag{8}$$

Equalized odds aims for outcome independence across demographic groups when conditioned on the true label (Hardt et al., 2016a).

These metrics, while essential for assessing fairness, have inherent trade-offs and limitations. It is generally impossible to satisfy all these conditions simultaneously when the base rates differ across groups (Chouldechova, 2017). They focus on ensuring parity in predictions without necessarily considering the impact on overall task accuracy (Dwork et al., 2012; Hardt et al., 2016b).

Recognizing the need to balance accuracy and fairness, there's a growing emphasis on metrics that integrate classification performance, creating more robust and equitable machine learning systems suitable for real-world applications Grgić-Hlača et al. (2018). This unified approach fosters a comprehensive evaluation and comparison of debiasing models, ensuring effectiveness in predictions while maintaining fairness.

## B  THE ADJUSTED PARITY METRIC

As an initial step towards solving these issues, introduced a parity metric for evaluating domain invariance (Vowels et al., 2020). This metric accommodates both discrepancies in accuracy across domains and normalised classifier or regressor performance to provide a single unified value for comparing debiasing models.

The adjusted parity metric was originally expressed for binary domains as:

$$\Delta_{\text{adj}} = \bar{S}(1 - 2\sigma), \tag{9}$$

where $\sigma$ represents the standard deviation of the normalised classifier accuracy across the domains, and $\bar{S}$ denotes the average accuracy over the domains.

We extend this definition to an arbitrary number of domains:

$$\Delta_{\text{adj}} = \frac{\bar{S} - S^R}{1 - S^R}\left(1 - \frac{\sigma}{\gamma}\right), \tag{10}$$

where $S^R$ is the baseline accuracy of a random predictor, and $\gamma$ is the maximum standard deviation across domains. This serves to normalise the metric between [0,1].

The introduction of $\gamma$ in this paper extends the metric to sensitive characteristics with more than two domains. For any even number of domains ($N$) the value remains at $\gamma = 0.5$ as in the original formulation. However for an odd number of domains:

$$\gamma = \sqrt{\frac{1}{4}\left(1 - \frac{1}{N^2}\right)}. \tag{11}$$

Please see the appendix for a full derivation.

The implications of this metric are twofold. Firstly, any classifier that exhibits either minimal consistency or accuracy will yield $\Delta_{\text{adj}} = 0$. Conversely, only a classifier that demonstrates maximal consistency and accuracy will result in $\Delta_{\text{adj}} = 1$. Figure 6 shows how $\Delta_{\text{adj}}$ changes with $\bar{S}$ for various $\sigma$. The motivation behind developing this metric stems from an essential understanding that invariance to a domain or attribute does not necessarily equate to reliable classification. A representation must also be informative for the intended task to be deemed effective.

## C    DIFFERENTIAL ADJUSTED PARITY: FULL FORMULATION

In order to propose a differentiable variant of the adjusted parity metric from equation 10, we must first rely on a differentiable variant of the accuracy measure $S$. In this paper, we propose the use of "Soft Balanced Accuracy". This is a differentiable form of balanced accuracy, which allows us to simultaneously deal with the challenges common in imbalanced problems.

The soft accuracy is computed by omitting the "argmax" function from a standard classification accuracy metric. In other words, for a vector of predicted class probabilities $P(x)$ given input $x$ and a one-hot encoded label vector $L_x$, the vector of soft True Positive rates for all classes is:

$$\text{TP} = \sum_x P(x) \odot L_x, \tag{12}$$

where $\odot$ represents the Hadamard product.

Analogously, the vector of Soft False Positives (FP) for each class would be computed as the sum of the predicted probabilities for instances that are incorrectly predicted as that. Given the inverted (one-cold) label vector $\bar{L}_x$:

$$\text{FP} = \sum_x P(x) \odot \bar{L}_x. \tag{13}$$

We can similarly compute the vectors of Soft True Negatives (TN) and Soft False Negatives (FN) as

$$\text{TN} = \sum_x (1 - P(x)) \odot \bar{L}_x, \tag{14}$$

$$\text{FN} = \sum_x (1 - P(x)) \odot L_x. \tag{15}$$

Although it may be obvious, it is worth pointing out that these soft variants of the TP, TN, FP, and FN are easily differentiable, as they are computed directly from the predicted probabilities. It is also worth pointing out that the classes referred to here are based on the output task, and differ from the sensitive characteristic domains of equation 11.

Given the above, we could compute the per-class accuracy vector as

$$\text{Acc} = \frac{(\text{TP} + \text{TN})}{(\text{TP} + \text{TN} + \text{FP} + \text{FN})}. \tag{16}$$

However, this measure can be misleading when classes are imbalanced. For instance, in a dataset where a single class represents 90% of the data, a naive classifier that always predicts the dominant class will have an accuracy of 90%.

To address this, we calculate the Soft Balanced Accuracy, which is the average of the per-class recall. In other words

$$S = \frac{1}{C} \left| \frac{\text{TP}}{(\text{TP} + \text{FN})} \right|^1, \tag{17}$$

where $C$ is the number of classes and $|.|^1$ represents the L1 norm. This gives equal weighting to all classes irrespective of their prevalence in the dataset.

To return to the definition of $\Delta_{adj}$ in Section B: we propose computing this soft balanced accuracy $S$ independently on subsets of the dataset corresponding to each sensitive domain. The mean and standard deviation of $S$ across these domains can then be substituted for $\bar{S}$ and $\sigma$ in equation 10 (once again noting that the set of task labels is not the same as the set of sensitive characteristics). By substituting the balanced soft accuracy into the adjusted parity metric, we can obtain a differential adjusted parity (DAP) loss which we can use to train a model. This loss encourages improvements in task accuracy across all labels, weighted by their prevalence and the current relative task performance, with no adversarial component. Intuitively, minimising task prediction inconsistency across sensitive domains would minimise the mutual information between the representations and the sensitive feature. This leads to less information regarding the sensitive feature being encoded in the representations and thus better demographic parity and equalised odds scores.

## D    DETAILED EVALUATION PROTOCOL

In our experiments, we've chosen to focus on two widely-acknowledged datasets in fairness research: the Adult dataset and the COMPAS dataset. In both cases we train a network using a combination of our DAP metric and the standard task cross-entropy loss ($\mathcal{L}_{ce}$). We also introduce weighting hyperparameters $\beta$ and $\Omega$ which control the contribution of standard deviation term and $\mathcal{L}_{ce}$ respectively.

### D.1    ADULT DATASET

The Adult dataset, often referred to as the "Census Income" dataset, originates from the UCI Machine Learning Repository (Dua & Graff, 2017). It comprises demographic data extracted from the 1994 Census Bureau database. The primary task for this dataset is binary classification: predicting whether an individual earns more than $50k annually based on attributes like age, occupation, education, and marital status.

One notable characteristic of the Adult dataset is its inherent imbalance. Specifically, a substantial proportion of individuals in the dataset have incomes below $50k (around 75.4%). The dataset contains several sensitive attributes such as race and gender. We opt to use gender as the sensitive feature in this evaluation. This is also imbalanced with roughly 67.3% of the data being male. Such imbalances could mislead naive classifiers into an unwanted bias towards the dominant class. Both the gender feature and target income variable are binary. We attempt to eliminate disparities in income predictions across gender groups.

### D.2    COMPAS DATASET

The Correctional Offender Management Profiling for Alternative Sanctions (COMPAS) dataset became notably popular following an investigation by ProPublica in 2016 (Dieterich et al., 2016). COMPAS is a risk assessment tool used in the U.S. legal system to assess the likelihood that a defendant will re-offend. Each instance in the dataset contains 12 features like age, gender, criminal history, and risk scores. The primary task is to predict if an individual will re-offend within two years.

ProPublica's analysis notably highlighted racial disparities in the predictions, where African-American defendants were more likely to be falsely classified as high risk compared to white defendants. We therefore opt to use race as the sensitive feature. The COMPAS dataset is balanced in terms of both the sensitive feature and target variable. The COMPAS dataset one-hot encodes ethnicity into five categories: African American, Asian, Hispanic, Native American, and Other. Studies often reduce this multi-class feature into a binary one distinguishing only between African-American and all other ethnicities, overshadowing the multi-class complexity. To enable comparison against models that do not have multi-class sensitive feature debiasing capability, we perform experiments with this binary simplification. However, we also evaluate our approach with the true multi-class problem. It is important to note that the COMPAS dataset has been heavily citricised for its use in fairness research due to its inherent measurement biases and errors, its disconnection from real-world criminal justice outcomes and its lack of consideration for the complex normative issues related to fairness, justice and equality (Bao et al., 2021). We use it here only to support comparison against previous works.

### D.3    DATA PREPROCESSING

For both datasets, we performed standard preprocessing, mapping categorical features to numerical indices, normalization of continuous variables, and handling missing values by replacing them with -1. We split the datasets into training and test sets in a 175:25 ratio. We also drop redundant features, that are either repeated or with mostly missing values.

### D.4    HYPERPARAMETER AND MODEL TRAINING

We employed a learning rate of 0.005 and 0.01 and a batch size of 64 and 32 for the Adult and COMPAS datasets respectively. The models were trained for 20 epochs.

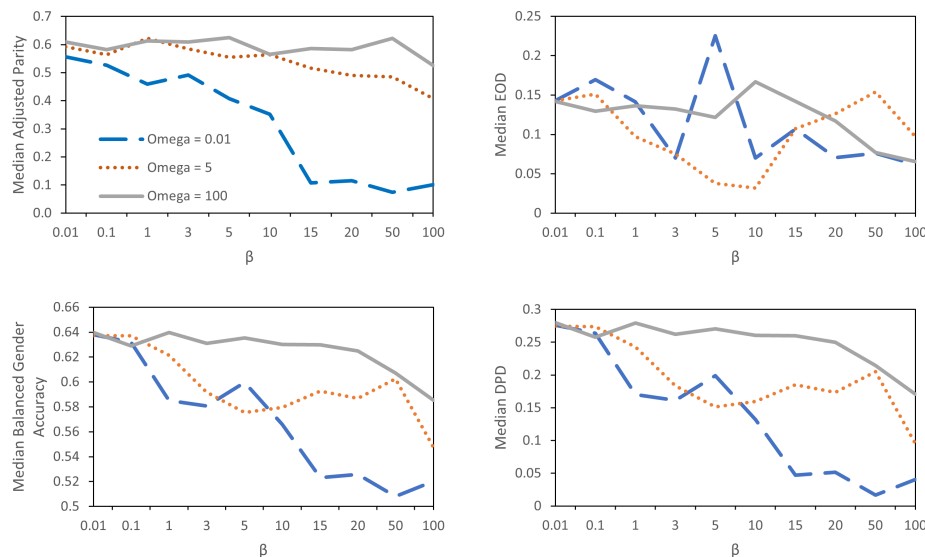

Figure 2: Effect of altering $\Omega$ and $\beta$ on adjusted parity (top-left), EOD (top-right), gender accuracy (bottom-left), and DPD (bottom-right) on Adult dataset. Higher adjusted parity and lower EOD, DPD and gender accuracy are favourable. Increasing $\beta$ lowers adjusted parity but improves all other metrics. Effect more pronounced at lower $\Omega$

For our hyperparameter sensitivity study, we chose values $\Omega \in \{0...100\}$ and $\beta \in \{0.1...100\}$, resulting in 100 tested combinations of $\Omega$ and $\beta$. All models were trained through 5 distinct runs, and we report the median as well as standard deviation of their performance across the runs. Given the stochastic nature of neural network training, this ensured robustness in our findings.

## D.5   EVALUATION FRAMEWORK

To evaluate our models' performance, we trained 2 balanced random forest classifiers to predict the sensitive and target features from the encodings in the testing phase. This allowed us to measure balanced classification accuracies on the task variable and sensitive feature from the embeddings produced during testing. On the fairness front, we obtain fairness metrics like demographic parity and equalised odds, using the predicted target from the random forest classifier. We also obtain an adjusted parity metric for comparing models and selecting best performing hyperparameters.

We compare our model with CFAIR, LAFTR and FAIR (ALFR), which we covered in Section A. We also compare against NODEBIAS which is reference network devoid of fairness constraints. The implementation for these models was adapted from (Kim, 2021). We evaluate them using the same evaluation protocol as for our model with balanced random forest classifier to obtain target and sensitive feature classification accuracy, DP, EO and adjusted parity metrics. We also contrast our DAP model using balanced soft accuracies against a variant using unbalanced soft accuracies.

## E   ADDITIONAL RESULTS

Figure 2 delves into the effects of tweaking $\beta$ and $\Omega$ on the performance of DAP. When $\Omega$ is held constant and $\beta$ is gradually increased, there is a notable decrease in the adjusted parity. Concurrently, other fairness metrics such as EOD, DPD, and gender classification balanced accuracy witness marked improvements. This effect is particularly evident at lower $\Omega$ values. Conversely, when $\beta$ remains static and $\Omega$ is increased, the outcomes typically include a surge in adjusted parity, gender classification balanced accuracy, and DPD metrics. Interestingly, EOD doesn't seem to follow a discernible trend in relation to changing $\Omega$ values. It's important to highlight that specific pairings of $\beta$ and $\Omega$ can optimize EOD values, indicating the delicate interplay between these parameters.

| DAP Model | $\Omega$ | $\beta$ | Adjusted Parity | EOD | DPD | Gender Accuracy (%) |
|---|---|---|---|---|---|---|
| Balanced | 100 | 5 | 0.632±0.014 | 0.126±0.075 | 0.265±0.028 | 60.9±7.2 |
| Balanced | 10 | 3 | 0.617±0.005 | 0.090±0.005 | 0.233±0.011 | 62.3±4.2 |
| Balanced | 1 | 1 | 0.593±0.008 | 0.052±0.052 | 0.185±0.016 | 59.7±5.5 |
| Balanced | 15 | 5 | 0.621±0.009 | 0.111±0.010 | 0.242±0.018 | 61.1±1.9 |
| Balanced | 3 | 1 | 0.622±0.006 | 0.113±0.038 | 0.244±0.011 | 59.3±4.4 |
| Unbalanced | 100 | 5 | 0.635±0.013 | 0.122±0.059 | 0.271±0.025 | 62.5±5.9 |
| Unbalanced | 10 | 3 | 0.618±0.011 | 0.096±0.020 | 0.236±0.022 | 61.4±3.3 |
| Unbalanced | 1 | 1 | 0.596±0.010 | 0.053±0.033 | 0.193±0.019 | 61.0±4.1 |
| Unbalanced | 15 | 5 | 0.615±0.010 | 0.094±0.009 | 0.230±0.020 | 59.9±3.8 |
| Unbalanced | 3 | 1 | 0.619±0.005 | 0.114±0.045 | 0.239±0.011 | 60.7±5.2 |

Table 1: Comparing the adjusted parity, EOD, DPD, gender accuracy when balanced and unbalanced accuracies are used during training. This table shows a few high performing combinations of $\beta$ and $\Omega$ on the Adult dataset. No significant difference is observed between Balanced and Unbalanced.

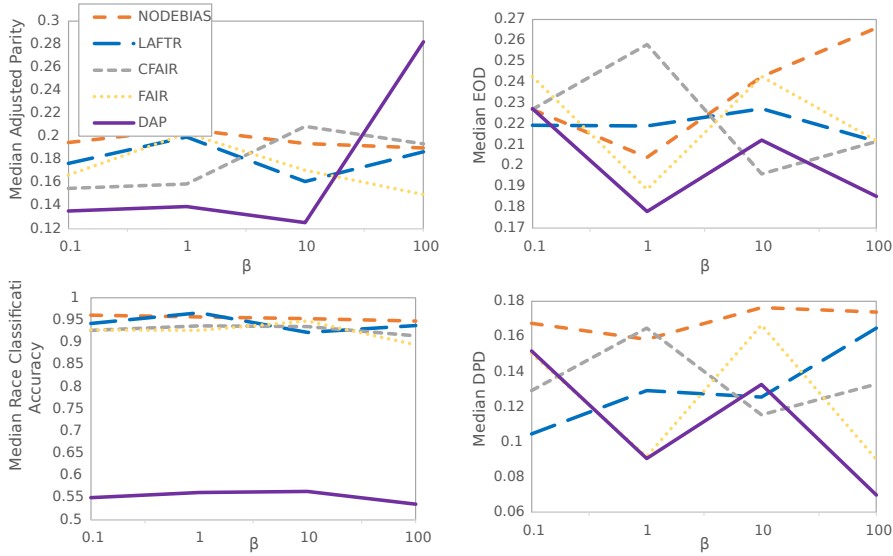

Figure 3: Comparing all 4 models performance and fairness against DAP on the COMPAS dataset. Top-left, top-right, bottom-left and bottom-right graph show how adjusted parity, EOD, race classification accuracy and DPD change with $\beta$. Higher adjusted parity and lower EOD, DPD and gender accuracy are favourable. DAP has lower gender classification accuracy at all $\beta$. Highest adjusted parity and lowest EOD and DPD are obtained by DAP

In an evaluative comparison between using balanced and unbalanced soft accuracies during training, Table 1 underscores that there are negligible differences in the adjusted parity. Moreover, the differences in adjusted parity, gender classification accuracy, EOD, and DPD are all insignificant as emphasized by the overlap of the standard deviations.

Figure 3 presents the experimental outcomes on the COMPAS dataset. Here DAP exhibits consistent improvements over previous state-of-the-art. In particular for high values of $\beta$ (meaning a high fairness weighting), DAP outperforms all competing approaches. DAP achieves an adjusted parity that is improved by 45.9% and an EOD that is reduced by 12.4% when compared with its nearest competitor, CFAIR. Similarly, it registers a 22.5% improvement in DPD and a substantial 40.1% reduction in race classification accuracy when compared to FAIR, the latter being the second-best performer for these metrics. Unlike with the Adult dataset, the performance of DAP on COMPAS does not seem to be very sensitive to the value of $\Omega$. Because the performance is roughly similar across all values, the results are omitted here and can be found in Section G. However, they can be found in the supplementary material.

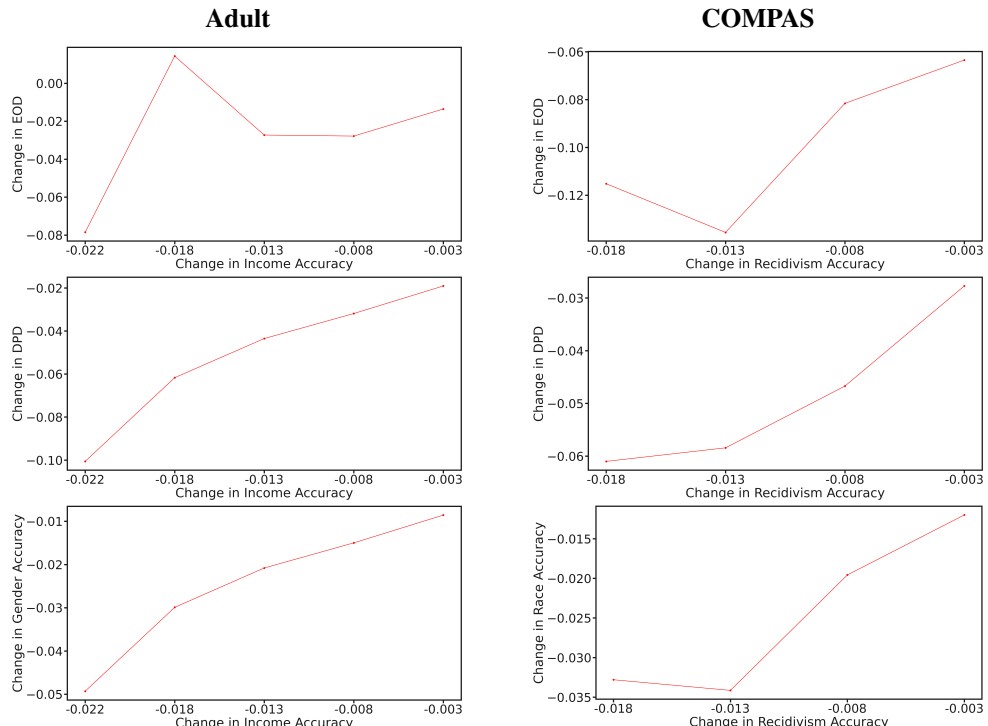

Figure 4: Showing how fairness metrics vary with changes in task accuracy from baseline values (Income Accuracy: 0.8294 for Adult, Recidivism Accuracy: 0.584 for COMPAS; EOD: 0.1429/0.310; DPD: 0.2759/0.173; Gender/Race Accuracy: 0.637/0.581) for each metric for Adult (left) and COMPAS (right). Changes were binned into 0.005 intervals, and averages for EOD, DPD, and gender/race accuracy were computed for each interval. Obtaining the largest negative change in EOD, DPD and sensitive feature accuracy for the least drop in task accuracy is favorable.

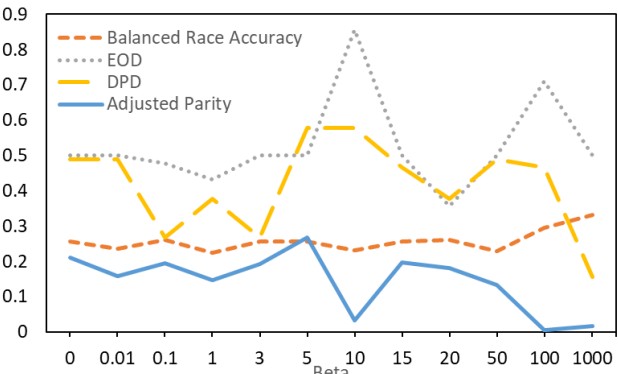

Figure 5: Demonstrating the performance of DAP with multi-class sensitive features at $\Omega$=20. DPD and adjusted parity decrease and race classification accuracy approaches 0.33 with increasing $\beta$, as desired. EOD shows no significant trend.

Figure 4 demonstrates the interplay between improving fairness metrics and declining task performance. Ideally, we aim to minimize the impact on task accuracy while maximizing the reduction in fairness-related metrics. With less than a 2.5% decrease in income classification accuracy, a reduction of 0.08, 0.10, and 4.93% is achieved on EOD, DPD, and gender classification accuracy, respectively, on the Adult dataset. Similarly, a decline of less than 2% in recidivism accuracy results in a decrease of 0.14, 0.06, and 3.41% in EOD, DPD, and race classification accuracy on the COMPAS dataset.

Figure 5 illustrates the effectiveness of DAP for multi-class sensitive attributes. As $\beta$ increases, EOD remains constant, starting and ending at 0.5 with no significant negative trend. In contrast, DPD significantly decreases by 68.18%, dropping from 0.4889 to 0.1556. Balanced race classification accuracy improves by 28.82%, rising from 0.2570 to 0.3311, nearing the random chance level of 0.33, indicating reduced race information in the embeddings. Finally, adjusted parity increased slightly up to 0.27 at $\beta = 5$, but overall shows a substantial drop of 92.14%, from 0.2100 to 0.0165.

## F  DERIVATION OF MAXIMUM STANDARD DEVIATION

We define our Differential Adjusted Parity as

$$\Delta_{\text{adj}} = (\bar{S} - S^r)\left(1 - \frac{\sigma}{\gamma}\right) \tag{18}$$

where $S^r$ is the baseline accuracy of a random predictor, $\bar{S}$ is the average Soft Balanced Accuracy across sensitive domains, $\sigma$ is the standard deviation of SAB across these domains and $\gamma$ is the maximum possible standard deviation. The term $\gamma$ serves to normalise the metric between [0,1].

For any even number of domains $(N)$ $\gamma = 0.5$. However for an odd number of domains

$$\gamma = \sqrt{\frac{1}{4}\left(1 - \frac{1}{N^2}\right)}. \tag{19}$$

Below we present the derivation of this rule.

The standard deviation of values, $\sigma$, is defined as:

$$\sigma = \sqrt{\frac{1}{N}\sum_{s \in S}(s - \bar{S})^2} \tag{20}$$

where $\bar{S}$ is the mean of the values in $S$, representing the mean of the soft balanced accuracies.

The formulation for the maximum standard deviation, $\gamma$, is:

$$\gamma = \arg\max_{S \in \mathbb{R}^N}\sqrt{\frac{1}{N}\sum_{s \in S}(s - \bar{S})^2} \qquad \text{s.t. } s \in [\alpha..\Omega] \tag{21}$$

where $\alpha$ and $\Omega$ are the upper and lower bounds on the values of $s$. In other words we use soft balanced accuracy vector $S$ which leads to maximal standard deviation from the mean.

We can see by inspection that when the values of $s$ are bounded, we achieve maximial standard deviation by making $\bar{S}$ as centered as possible within the range, while all the values in $s$ are on the extremes of the range. When $N$ is even, this can be perfectly achieved by placing half of the items at each end. In such a scenario, $\bar{S} = \sigma = \frac{\alpha+\Omega}{2}$. Hence, if the soft balanced accuracies are bounded between [0..1] then the maximum standard deviation for any even number of sensitive domains is 0.5.

However, for odd $N$, the mean will necessarily be slightly off-center from the range, leading to a lower maximal $\gamma$. Considering that $\text{floor}(N/2)$ items are placed at $\alpha$ and $\text{ceil}(N/2)$ items are placed at $\Omega$, the mean can be derived as:

$$\bar{S} = \frac{\alpha\text{floor}\left(\frac{N}{2}\right) + \Omega\text{ceil}\left(\frac{N}{2}\right)}{N} \tag{22}$$

$$= \frac{\alpha\frac{N-1}{2} + \Omega\frac{N+1}{2}}{N} \tag{23}$$

Similarly, the summation inside the definition of $\sigma$ can be split into 2 parts and resolved

$$\gamma = \sqrt{\frac{1}{N}\left(\sum_{i=1}^{\frac{N-1}{2}}(\bar{S} - \alpha)^2 + \sum_{i=1}^{\frac{N+1}{2}}(\bar{S} - \omega)^2\right)} \tag{24}$$

$$= \sqrt{\frac{1}{N}\left(\frac{N-1}{2}(\bar{S} - \alpha)^2 + \frac{N+1}{2}(\bar{S} - \omega)^2\right)} \tag{25}$$

Given that the minimum normalized accuracy, $\alpha$, is 0 and the maximum normalized accuracy, $\Omega$, is 1, the simplified mean is:

$$\bar{S} = \frac{1 + \frac{1}{N}}{2} \tag{26}$$

$$= \frac{1}{2} + \frac{1}{2N} \tag{27}$$

Consequently, we note that as $N \to \infty$, $\bar{S}$ approaches 0.5.

Similarly, substituting $\alpha = 0$, $\Omega = 1$ into equation 25 gives

$$\gamma = \sqrt{\frac{1}{N} \left( \frac{N-1}{2}(\bar{S}-0)^2 + \frac{N+1}{2}(\bar{S}-1)^2 \right)} \tag{28}$$

$$= \sqrt{\frac{1}{N} \left( \frac{N-1}{2}\bar{S}^2 + \frac{N+1}{2}(\bar{S}^2 - 2\bar{S} + 1) \right)} \tag{29}$$

$$= \sqrt{\left( \frac{1 - \frac{1}{N}}{2}\bar{S}^2 + \frac{1 + \frac{1}{N}}{2}(\bar{S}^2 - 2\bar{S} + 1) \right)} \tag{30}$$

$$= \sqrt{\left( \frac{\bar{S}^2}{2} - \frac{\bar{S}^2}{2N} \right) + \left( \frac{1}{2} + \frac{1}{2N} \right)(\bar{S}^2 - 2\bar{S} + 1)} \tag{31}$$

$$= \sqrt{\left( \frac{\bar{S}^2}{2} - \frac{\bar{S}^2}{2N} \right) + \left( \frac{\bar{S}^2}{2} - \bar{S} + \frac{1}{2} \right) + \left( \frac{\bar{S}^2}{2N} - \frac{\bar{S}}{N} + \frac{1}{2N} \right)} \tag{32}$$

$$= \sqrt{\bar{S}^2 \left( \frac{1}{2} - \frac{1}{2N} + \frac{1}{2} + \frac{1}{2N} \right) - \bar{S} \left( 1 + \frac{1}{N} \right) + \left( \frac{1}{2} + \frac{1}{2N} \right)} \tag{33}$$

$$= \sqrt{\bar{S}^2 - \bar{S} - \frac{\bar{S}}{N} + \frac{1}{2} + \frac{1}{2N}} \tag{34}$$

Finally, substituting equation 27 into equation 34 we can fully simplify:

$$\gamma = \sqrt{\left( \frac{1}{2} + \frac{1}{2N} \right)^2 - \left( \frac{1}{2} + \frac{1}{2N} \right) - \frac{\left( \frac{1}{2} + \frac{1}{2N} \right)}{N} + \frac{1}{2} + \frac{1}{2N}} \tag{35}$$

$$= \sqrt{\left( \frac{1}{4} + \frac{1}{4N^2} + \frac{1}{2N} \right) - \left( \frac{1}{2} + \frac{1}{2N} \right) - \left( \frac{1}{2N} + \frac{1}{2N^2} \right) + \frac{1}{2} + \frac{1}{2N}} \tag{36}$$

$$= \sqrt{\frac{1}{N^2} \left( \frac{1}{4} - \frac{1}{2} \right) + \frac{1}{4}} \tag{37}$$

$$= \sqrt{\frac{1}{4} \left( 1 - \frac{1}{N^2} \right)} \tag{38}$$

## G  HYPERPARAMETER SENSITIVITY ON THE COMPAS DATASET

Figure 7 shows the sensitivity of the DAP system to the $\beta$ and $\Omega$ hyperparameters on the COMPAS dataset. The results do not show the same trend witnessed on the Adult dataset with varying $\beta$ and $\Omega$. Increasing $\beta$ for a given $\Omega$ does not lower adjusted parity and improve fairness metrics as observed on the Adult dataset. The full set of results used to obtain graphs here and in the anonymous submission are placed with the zip file containing the code.

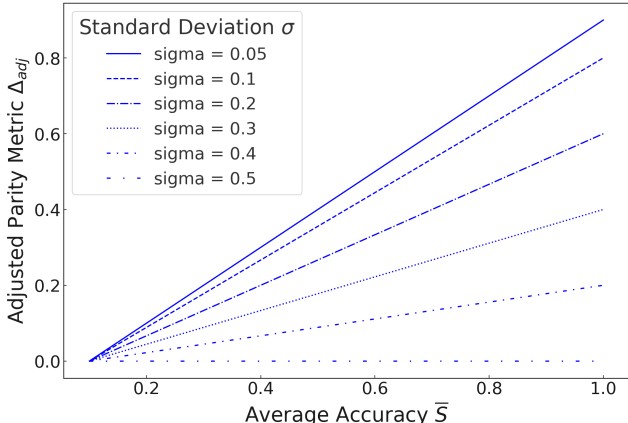

Figure 6: Adjusted Parity Metric $\Delta_{\text{adj}}$ as a function of the Average Accuracy $\overline{S}$ for various values of the Standard Deviation $\sigma$ across domains. The baseline accuracy of a random predictor $S^R$ is set to 0.1, and the maximum standard deviation $\gamma$ is fixed at 0.5. The metric demonstrates how increasing performance inconsistency (higher $\sigma$) across domains reduces $\Delta_{\text{adj}}$, even when average accuracy is high.

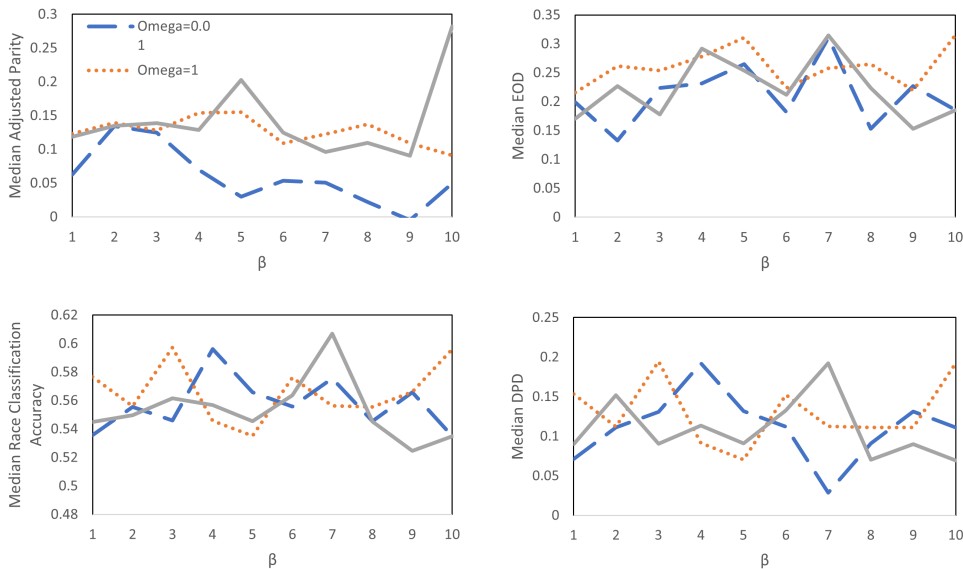

Figure 7: Effect of altering $\Omega$ and $\beta$ on adjusted parity (top-left), EOD (top-right), gender accuracy (bottom-left), and DPD (bottom-right) on COMPAS dataset. Higher adjusted parity and lower EOD, DPD and gender accuracy are favourable.

