# OpenReview forum: "Differential Adjusted Parity for Learning Fair Representations"
_ICLR.cc/2026/Workshop/AFAA — AFAA 2026 Poster_

### Official Review · Reviewer_21MY · 2026-02-19
**Hard to ascertain the significance of the contribution of the work**

**Rating:** 2
**Confidence:** 4

**Summary:**

The paper introduces an extension of the adjusted parity metric that it would work for multiple sensitive groups. Furthermore this expression is then transformed that it is differentiable and can be used in the training of the model. The paper then conducts some experiments on the bank dataset for the binary sensitive attribute of gender and on the COMPAS dataset on the categorical sensitive attribute of race, although a set of the experiments are conducted on the binary simplification.

**Strengths:**

Incorporating the fairness constraint directly into the training of the model is an intuitive way of enforcing fairness and provides more flexibility compared to models proposed 4 years ago.

The derivation of the new adjusted parity metric is mathematically derived.

The paper reports performance on a set of fairness constraints instead of just one as is often done.

**Weaknesses:**

The related work section is severely lacking other bias mitigation approaches. The focus of this work on adversarial debiasing is also strange as the proposed method does not fall under that category and adversarial debiasing has not been the standard for quite some time. Examples of methods that could/should have been discussed Celis et al., Buyl et al., Cruz & Hardt, and Delaney et al. Besides research papers mentioning one bias mitigation method, there are also two well-known toolkits, AIF360 and Fairlearn, which should have been mentioned. Additionally, in order to create a more fair comparison some of these methods should have been applied as to compare the method with more novel bias mitigation methods.

The paper does a poor job conveying the need for the adjusted parity metric. The questionable use of the metric is further strengthened by the limited uptake the original paper suggesting the metric saw.

The paper itself states the COMPAS dataset is problematic for determining bias mitigation capabilities of a method and yet is still one of their main datasets. The other dataset used is the Adult dataset which has received backlash too (Ding et al.) and alternatives for it have been proposed. Therefore, the paper uses no appropriate datasets for its task.

The statement made about combining demographic parity and equalized odds is false. The combination of both is possible, but would result in a trivial classifier. See the fair ML book and Defrance & De Bie.


The paper still contains several typos throughout the work, such as componenet in the third paragraph of the introduction.



L. Elisa Celis, Lingxiao Huang, Vijay Keswani, and Nisheeth K. Vishnoi. 2019. Classification with Fairness Constraints: A Meta-Algorithm with Provable Guarantees. FAT* '19.

Maarten Buyl, MaryBeth Defrance, Tijl De Bie. 2024. fairret: a Framework for Differentiable Fairness Regularization Terms. ICLR 2024.
André F. Cruz, Moritz Hardt. Unprocessing Seven Years of Algorithmic Fairness. ICLR 2024.

Eoin Delaney, Zihao Fu, Sandra Wachter, Brent Mittelstadt, Chris Russell. 2024. OxonFair: A Flexible Toolkit for Algorithmic Fairness. NeurIPS 2024.

Frances Ding, Moritz Hardt, John Miller, and Ludwig Schmidt. 2021. Retiring adult: new datasets for fair machine learning. (NIPS '21).

MaryBeth Defrance, Tijl De Bie. 2025. Maximal Combinations of Fairness Definitions. JAIR

---

### Official Review · Reviewer_h9nw · 2026-02-20
**Differentiable Fairness Metric with Promising Results but Limited Comparisons**

**Rating:** 4
**Confidence:** 3

**Summary:**

The paper proposes a differentiable fairness metric that simultaneously improves representation fairness and task accuracy. The authors first introduce a modified version of the adjusted parity metric and then derive a differentiable adjusted parity (DAP) formulation, in which both the mean accuracy and the standard deviation of accuracy across domains (attributes) are computed using a differentiable soft-balanced accuracy. This soft-balanced accuracy is computed from true positives (TP) and false negatives (FN) directly using predicted softmax probabilities, making the entire metric differentiable and suitable for optimization.

The method is evaluated on the Adult and COMPAS datasets, comparing the proposed objective with adjusted parity, classical fairness metrics, and existing adversarial-based approaches. Experimental results indicate that the proposed loss achieves better fairness–accuracy trade-offs than adversarial baselines while being more stable.

**Strengths:**

- The proposed method is effective at improving both adjusted parity and standard fairness metrics such as equality of odds and demographic parity.

- The formulation of differentiable true positives and false negatives based on predicted probabilities is interesting and potentially impactful, as it may enable the design of additional differentiable fairness metrics.

- The paper is generally well written, and the experimental setup is clearly described.

**Weaknesses:**

1. **Motivation and novelty of the metric.**
   The motivation illustrated in Figure 6 (i.e., that larger variance across domains decreases the metric value) already holds for the original adjusted parity formulation. It would be helpful to more clearly articulate the additional value the proposed modification provides. Why not make the original adjusted parity differentiable? How does it compare?

2. **Comparisons limited to adversarial methods.**
   The empirical evaluation only compares against adversarial debiasing approaches. Including comparisons with non-adversarial debiasing methods mentioned in section 2.2 would strengthen the experimental section and provide a more complete assessment.

3. **Undefined abbreviations.**
   Several abbreviations appear without prior definition, which reduces readability:
   - FRC (Line 81), BFA (Line 84)
   - EOD and DPD in the experimental section
   Please define all acronyms at first occurrence and/or give proper reference.

4. **Clarity in Section 2.2 (Lines 77–83).**
   The paragraph could be clearer (cf. previous point). How is $H(z)$, the entropy of the representations z, defined? Is this the entropy of a predicted distribution $P$, or is it computed directly from representation vectors?

---

### Official Review · Reviewer_Q2xe · 2026-02-23
**A non-adversarial approach to increase Fairness-accuracy trade-off and able to handle several sensitive domains at once: very promising results, need of few more architecture details, looking forward to a principled way of choosing key hyperparameters for the loss.**

**Rating:** 4
**Confidence:** 2

**Summary:**

The paper proposes a non-adversarial loss function for learning fair machine learning representations by employing Soft Balanced Accuracy to make the adjusted parity metric  fully differentiable. Experiments are performed on the Adult (income prediction, gender-sensitive) and COMPAS (recidivism prediction, race-sensitive) datasets, and compared against ALFR, CFAIR, LAFTR, and a  reference network devoid of fairness constraints.

**Strengths:**

1-) The paper reads easily
2-) Presence of previous/related works
3-) Experimental results show very good improvements on fairness  with only small drops in task accuracy
4-) No need of adversarial training
5-) The method can handle multiclass sensitive features.

**Weaknesses:**

1-) Few typos
2-) The ADP loss seems not to be explicitly written?
3-) I may be wrong, but it seems to me that comparisons against other non-adversarial debiasing are missing?
4-) It would be good to have more details about the network trained for experiments.
5-) As the authors mentioned, it would be nice to have indications in future works how to choose the \beta and \Omega parameters.

---

### Meta-Review · Area_Chair_WSQL · 2026-02-26

**Recommendation:** Reject
**Confidence:** 3

**Metareview:**

The paper introduces Differential Adjusted Parity (DAP), a non-adversarial loss function designed to produce fair representations by utilizing a differentiable variant of the adjusted parity metric. The core contribution lies in the use of "soft balanced accuracies" to create a unified objective function that balances task performance and fairness without the instability often associated with adversarial training.

**Strengths:**

* The proposed approach is conceptually interesting and simplifies the debiasing process by avoiding adversarial components.

* It demonstrates the ability to handle multi-class sensitive features, a notable extension over many existing binary-focused methods.

* Preliminary results on Adult and COMPAS datasets show improvements in fairness metrics like demographic parity and equalized odds compared to some adversarial baselines.


**Weaknesses:**

* **Experimental Depth:** As pointed out by reviewer 21MY and echoed by Q2xe and h9nw, the experimental evaluation is limited to only two datasets. Furthermore, the paper lacks comparisons with contemporary non-adversarial algorithms of the same family, leaving an "experimental vacuum" regarding its relative state-of-the-art performance. I might missed something, but the multi-class scenario is not expored even being a core contribution of the paper.
* **Clarity and Motivation:** The introduction is underdeveloped and fails to sufficiently contextualize the literature or strongly motivate the specific necessity of the DAP loss over existing smoothing techniques.
* **Related Work:** The discussion of prior work is poorly articulated, lacking a critical synthesis that would help position this contribution within the broader field of fair representation learning.

Given these shortcomings in execution and evaluation, I believe that this paper should be accepted given the early-stage regarding experiments. However, the authors are strongly encouraged to address the reviewers' concerns regarding the lack of relevant baselines and to significantly refine the introduction and related work sections to meet the expected standards of clarity and scholarly rigor.

---

### Decision · Program_Chairs · 2026-03-02

**Decision:**

Accept (Poster)

**Comment:**

The paper was originally submitted under the Main Track. While the paper is not ready to be accepted as a Main Track Paper, we find the work promising, and are giving an opportunity to the authors to instead get accepted in the Tiny/Short Track. Please submit a camera-ready version of up to 3 pages to comply with the Tiny/Short Paper requirements (more instructions on how to submit the camera-ready version will follow soon). The authors can also decide to withdraw if they prefer to not be accepted under the Tiny/Short Track.